# Study on the Distribution of Low Molecular Weight Metabolites in Mango Fruit by Air Flow-Assisted Ionization Mass Spectrometry Imaging

**DOI:** 10.3390/molecules27185873

**Published:** 2022-09-10

**Authors:** Deqing Zhao, Ping Yu, Bingjun Han, Fei Qiao

**Affiliations:** 1Key Laboratory of Crop Gene Resources and Germplasm Enhancement in Southern China, Ministry of Agriculture/Tropical Crops Genetic Resources Institute, Chinese Academy of Tropical Agricultural Sciences, Haikou 571101, China; 2Yunyang County Agricultural Technology Service Center of Chongqing, Chongqing 404500, China; 3Key Laboratory of Quality and Safety Control for Subtropical Fruit and Vegetable, Ministry of Agriculture and Rural Affairs, Hainan Provincial Key Laboratory of Quality and Safety for Tropical Fruits and Vegetables, Analysis and Test Center, Chinese Academy of Tropical Agricultural Sciences, Haikou 571101, China

**Keywords:** mango, tissue, spatial distribution, mass spectrometry imaging

## Abstract

Mass spectrometry imaging is a novel molecular imaging technique that has been developing rapidly in recent years. Air flow-assisted ionization mass spectrometry imaging (AFAI-MSI) has received wide attention in the biomedical field because of its features such as not needing a pretreatment sample, having high sensitivity, and wide coverage of metabolite detection. In this study, we set up a mass spectrometry imaging method for analyzing low molecular metabolites in mango fruits by the AFAI-MSI method. Compounds such as organic acids, vitamin C, and phenols were detected from mango tissue by mass spectrometry under the negative ion scanning mode, and their spatial distribution was analyzed. As a result, all the target compounds showed different distributions. Citric acid was mainly distributed in the pulp. Malic acid, quinic acid, and vitamin C universally existed in the pulp and peel. However, galloylglucose isomer and 5-galloylquinic acid were predominantly found in the peel. These results show that AFAI-MSI can be used for the analysis of mango fruit endogenous metabolites conveniently and directly, which will facilitate the rapid identification and in situ characterization of plant endogenous substances.

## 1. Introduction

Mass spectrometry imaging (MSI) makes it possible to visualize distribution information of atoms and molecules on a sample surface. There are three representive MSI techniques: secondary ion mass spectrometry (SIMS) and matrix-assisted laser desorption ionization (MALDI) mass spectrometry imaging, which both require ionization under vacuum conditions, as well as desorption electrospray ionization (DESI) mass spectrometry imaging, an open ionization mass spectrometry imaging technology [1,2,3,4]. Initially, SIMS was used to analyze the surface of inorganic materials, but it has been extended to imaging low molecular weight materials on the surface of living tissues since the 1990s [5]. Meanwhile, MALDI-MSI was developed for proteins, and then used for macromolecules such as proteins and peptides [6,7]. As new matrixes are developed, MALDI-MSI is now widely used for endogenous small molecule imaging. For example, Sun et al. [8] used a simple acetone washing method to improve the sensitivity of MALDI-MS for small molecule metabolites, including polyamines, cholines, carnitines, amino acids, nitrogenous bases, nucleosides, carbohydrates, organic acids, and vitamins.

In recent years, ion sources which can ionize at atmospheric pressure have become increasingly popular since they have the advantages of operating in an open environment, being convenient to use, with no matrix needed, being low- or noninvasive, etc. [9]. Therein, air flow-assisted ionization mass spectrometry imaging (AFAI-MSI) was developed, and has been playing vital roles in drug metabolism research and clinical molecular histopathological diagnosis. Examples include the application of AFAI-MSI techniques to study the overall distribution of s-(+)-deoxytylophorinidine in rats, metabolite profiling of nasopharyngeal tissue specimens (nasopharyngeal carcinoma and chronic inflammation of nasal mucosa), and mass spectrometry imaging of endogenous metabolites in rat kidney tissue [10,11,12]. MSI technology has also brought great hope for the spatio-temporal analysis of plant tissues. Since 2005, MSI has been used to measure the spatial distribution of plant metabolites; for example, it has been used to identify precursors or related metabolites in order to figure out how plants react to stress, and to speculate new metabolic pathways [13].

Mango *(Mangifera indica* L.), belonging to family *Anacardiaceae*, is cultured all over the world in tropical and subtropical regions. Mango fruit is very popular for its attractive flavor, and can be consumed from the immature stage to the fully ripened state. Unripened fruit can be processed into pickle, chutney, mango sauce, raw mango powder (*amchoor*), and green mango drink (*panna*), while mature fruit is used to produce pulp, squash, nectar, beverages, mango leather (*Amb Papad*), mango puree, mango fruit bars, frozen and canned mango slices, and jam [14]. The mango fruit provides a significant source of macronutrients including carbohydrates, lipids and fatty acids, proteins and amino acids, and organic acids. In addition, mango contains micronutrients such as vitamins and minerals [15]. Most of the compounds in mango have been identified by the combination of analytical mass spectrometry (MS) and separation techniques such as gas chromatography (GC) and liquid chromatography (LC) [16,17]. Using these techniques, samples must be homogenized and extracted before being analyzed, and therefore the spatial distribution of compounds in fruits cannot be determined. In this study, the spatial distribution of low molecular weight metabolites in mango fruit was explored by AFAI-MSI. This will shed light on revealing the spatio-temporal distribution of low molecular metabolites in mango fruit.

## 2. Results and Discussion

Because of the integrity of the sections, the tissue freezing medium, the scanning mode of the mass spectrometry, and the type of spray solvent had a significant influence on the imaging results; therefore, these conditions are optimized by a series of experiments before mass spectrometry imaging. In brief, it was found that the mango tissue should not be embedded, but a little tissue freeze medium added to the surface of the freezing tray to fix the sample was helpful for slicing. Then, the mango tissue was sliced into 100 μm in thickness, and 80% methanol solution (contain 0.1% ammonia) was selected as the spray solvent for scanning analysis in the negative ion mode. In order to directly observe and reveal the spatial distribution of metabolites in mango, we also observed the microstructure of the tissue section and spliced the picture into a complete figure by using the ‘MosaiX Acquisition’ function in the software AxioVision (Figure 1).

The low molecular weight metabolites of mangoes determined by using AFAI-MSI were displayed in Table 1. A representative AFAI-MSI spectrum obtained from the strongest responses of each substance in mango tissue sections is depicted in Figure 2. The deprotonated molecules [M-H]^−^ of citric acid (*m*/*z* 191.0192) and malic acid (*m*/*z* 133.0133) both have a strong intensity, which is around 4.8 × 10^5^ and 2.7 × 10^5^ (Figure 2A,B), respectively. Next is vitamin C (*m*/*z* 175.0241), with an intensity of 1.1 × 10^5^ (Figure 2C). A weaker response is observed for quinic acid (*m*/*z* 191.0556) (Figure 2D), with an intensity of 2 × 10^4^. Lastly, there are 5-galloylquinic acid (*m*/*z* 343.0676) and galloylglucose isomer (*m*/*z* 331.0675) (Figure 2E,F), which had the weakest intensity of 2 × 10^4^.

Organic acids are essential for aerobic metabolism and serve as flavoring agents that affect fruit acidity and organoleptic qualities [18]. Mango fruit acidity is primarily attributable to citric and malic acids, although other common organic acids from the tricarboxylic acid cycle have been reported in mango fruit, such as citric, oxalic, succinic, malic, pyruvic, as well as tartaric, muconic, galipic, glucuronic, and galacturonic acids, whereas citric is the most abundant one [19,20,21,22]. Interestingly, in this study, it was found that citric acid was mainly located in the pulp, especially in the tissues closer to the seeds in all samples. However, the presence of citric acid was hardly detected in the peel and subcutaneous tissues (Figure 3A). In contrast, malic acid was distributed throughout the mango slices, with only a few places in the subcutaneous tissue showing significant amounts (Figure 3B). Pierson et al. [23] investigated the content and distribution of phytochemicals in mango peel and pulp, where the citric acid was detected by HPLC and HPLC-MS, and which was principally identified in the flesh of H10, Irwin, and Kensington Pride. The same compound was also present in the peel of the H10 variety at a lower level than the corresponding flesh. The above results indicate that the distribution of citric acid is varied among mango varieties.

Vitamin C (L-ascorbic acid), an antioxidant and immune booster, is necessary for collagen repair, prevention of scurvy, and absorption of iron [24]. As a vitamin C-rich fruit, its localization was also investigated by HPLC, and it showed that vitamin C is located in the whole mango tissue, including the pulp and peel. However, with the AFAI-MSI technique, we were able to further point out that the highest concentration is in the subcutaneous tissue (Figure 3C), which is in accordance with the phenomenon that vitamin C is preferentially located in photosynthetic cells and meristems [25]. Moreover, the higher vitamin C content in subcutaneous cells may also benefit the delaying of fruit senescence and defence against fruit diseases [26,27,28].

Quinic acid is a naturally occurring cyclohexanecarboxylic acid found in both plants and bacteria [29]. Here, the quinic acid is mainly found in the mango peel and subcutaneous tissue, which is nearly superposed with the distribution of vitamin C (Figure 3D). Related studies have shown that quinic acid is a by-product located in chloroplasts, and is produced by 3-dehydroquinic acid via the shikimate acid pathway, coupled with photosynthesis to regulate the biosynthesis of aromatic compounds in chloroplasts [30,31,32]. According to the microstructure of mango in Figure 1, there are many secretory cavities of various sizes in the mango subcutaneous tissue, which have been suggested as the primary location for storing aromatic compounds [33]. Therefore, we speculate that the higher content of quinic acid in the subcutaneous tissue of mango may be involved in the biosynthesis of aromatic compounds.

Phenolic acids are plant secondary metabolites that are important in the human diet owing to their biological activities and health benefits [34,35]. Mango contains two major classes of phenolic acids in plants: hydroxybenzoic acid and hydroxycinnamic acid derivatives, which can exist in free or conjugated form with glucose or quinic acid [36,37]. Gómez-Caravaca et al. [38] determined the presence of phenolic and other polar compounds in the edible part of mango and its by-products (peel, seed, and seed husk). They found that 5-galloylquinic acid and galloylglucose isomers were only detected in mango peel. In our case, two phenolic acid derivatives, 5-galloylquinic acid and galloylglucose isomer, could be detected, respectively (Figure 3E,F). 5-Galloylquinic acid and galloylglucose isomer are mainly distributed in the subcutaneous tissue of mango and can hardly be found anywhere else.

## 3. Materials and Methods

### 3.1. Plant Material

Fruits of *Mangifera indica* L. var. ‘Guifei’ were used in this study. The mango fruits were harvested from a local orchard in Dongfang, China. Fruit at the mature green stage had total soluble solids (TSS) of around 8%, and titratable acidity (TA) was around 1%. Mango color parameters were 65.82 (L*), −7.50 (a*), 9.26 (b*), respectively. Fresh fruit of moderate size and with an undamaged surface was selected as plant materials. The mango was cut into small pieces (about 9 mm × 9 mm), and fixed on the cryogenic platform using about 1 mL of OCT freezing medium. The samples were then mounted on a frozen tray, frozen for 30 min at −20 °C, and then sliced at a thickness of 100 µm by a cryostat microtome (LEICA XM1950UV, Leica Microsystem, Wetzlar, Germany). The microstructure was observed and photographed with a microscope (ZEISS Axio Observer Z1, ZEISS, Oberkochen, Germany) at a magnification of 100 times, before mass spectrometry imaging was performed. In this work, the experiment was repeated three times, which were named slice-1, slice-2, and slice-3, respectively.

### 3.2. AFAI-MS Imaging

The measurement was carried out using an AFAI ion source coupled to a Q-Orbitrap mass spectrometer (Q Exactive Plus, Thermo Scientific, Waltham, MA, USA). The mass spectra were acquired under the negative mode, with a scan range of 70–1000 Da, a mass resolution of 70,000, an automatic gain control target of 3 × 10^6^, and a maximum injection time of 200 ms. The spray voltage and transport tube voltage were set to 2 and 3.2 kV, respectively. N_2_ as spray gas was set to 0.6 MPa, and 80% methanol solution (*v*/*v*, contain 0.1% ammonia) was used as spray solvent. The spray solvent flow rate was 6 μL/min, and the air as assisting gas was set to 40 L/min. Xcalibur software (Version 2.2, Thermo Scientific, Waltham, MA, USA) was used to collect the data.

### 3.3. Data Processing

The original data file was converted into CDF format by Xcalibur software (Version 2.2, Thermo Fisher Scientific, San Jose, CA, USA), and then the file was read by mass spectrometry imaging software MassImager (MSI system workstation version 1.0, Beijing, China) to detect the type, relative intensity, and spatial position of ions for imaging analysis.

## 4. Conclusions

In this work, we applied AFAI-MSI to mango fruit research for the first time. Using this procedure, target compounds can be imaged in mango within a short period of time without the need for a specific chemical matrix and specific probes. Meanwhile, the technique was used to reveal the distribution of some major compounds in mango fruits. The distribution of target compounds in mango fruits is uneven, but all the compounds we analyzed showed the same trend in the location of the repeated samples. Citric acid was located in the pulp, while malic acid was evenly distributed in the whole mango tissue. Vitamin C and quinic acid were mainly distributed in the peel and subcutaneous tissue. 5-galloylquinic acid and galloylglucose isomer were found only in the subcutaneous tissues. This newly established method for mango flesh bioimaging broadens the application area of AFAI-MSI, and also contributes to gaining more insight into the biological or physiological functions of plant secondary metabolites, the identification of biomarkers, the process of metabolites biosynthesis, and the transportation of metabolites under biotic or abiotic stresses.

## Figures and Tables

**Figure 1 molecules-27-05873-f001:**
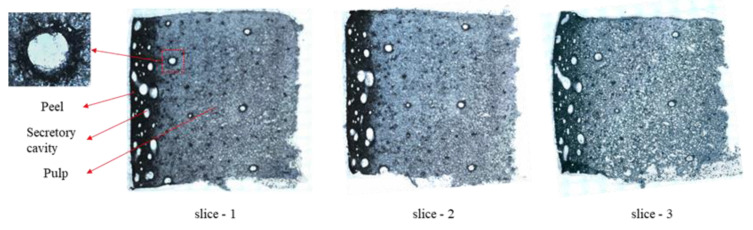
Microstructure of mango tissue slice, ×100.

**Figure 2 molecules-27-05873-f002:**
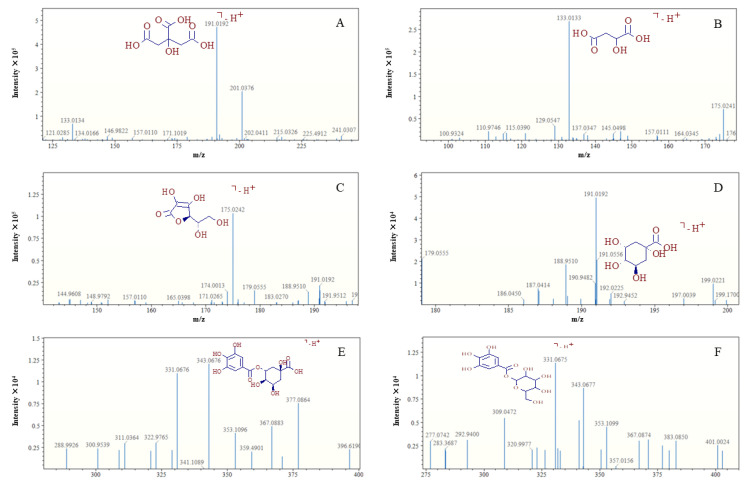
Representative AFAI-MSI spectra: citric acid (**A**), malic acid (**B**), vitamin C (**C**), quinic acid (**D**), 5-galloylquinic acid (**E**), galloylglucose isomer (**F**).

**Figure 3 molecules-27-05873-f003:**
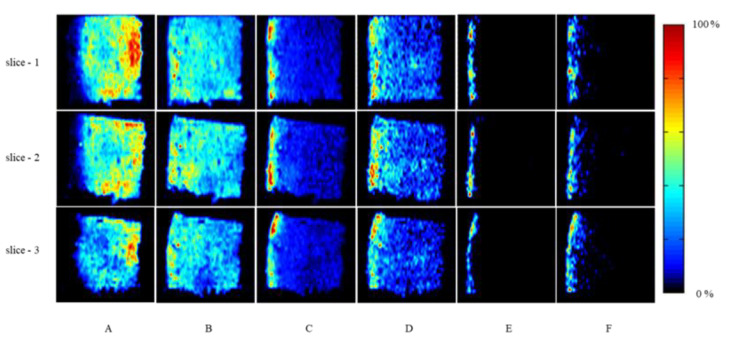
AFAI-MSI images of low molecular metabolites in mango: citric acid (**A**), malic acid (**B**), vitamin C (**C**), quinic acid (**D**), 5-galloylquinic acid (**E**), galloylglucose isomer (**F**). The color bar represents the relative intensity of compound ions.

**Table 1 molecules-27-05873-t001:** Ion formula, theoretical mass, and observed mass of low molecular metabolites in mango. The mass difference was represented as millidalton (mDa).

Compound Name	Ion Formula	Theoretical Mass (Da)	Observed Mass (Da)	Mass Difference (mDa)
Citric acid	[C6H8O7-H]^−^	191.0197	191.0192	−0.5
Malic acid	[C4H6O5-H]^−^	133.0142	133.0133	−0.9
Vitamin C	[C6H8O6-H]^−^	175.0248	175.0242	−0.6
Quinic acid	[C7H12O6-H]^−^	191.0561	191.0556	−0.5
5-Galloylquinic acid	[C14H16O10-H]^−^	343.0670	343.0676	0.6
Galloylglucose isomer	[C13H16O10-H]^−^	331.0670	331.0675	0.5

## Data Availability

The data presented in this study are available on request from the corresponding author.

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
