# Peer review of "Study on the Distribution of Low Molecular Weight Metabolites in Mango Fruit by Air Flow-Assisted Ionization Mass Spectrometry Imaging"

_molecules, 2022, doi:10.3390/molecules27185873_

Round 1
Reviewer 1 Report
The authors propose a new method to detect several compounds in Mango fruits using MS. The method is interesting and could have application to other matrices. Additionally, the authors described potentially relevant findings.
However, the experimental procedure was described vaguely, without objective parameters being provided for the selection of which fruits would be analyzed (e.g. solid content, acidity, RGB color measurements, etc), no quantification of the amount of freezing agent used, among others.
Also, the results are presented superficially, without information about reproducibility, inter sample variation, etc.
For this reason, I'd recommend the authors to rework the paper, including detailed information, and re-submit the paper.
Reviewer 2 Report
The authors has done great work in manuscript entitled as” Study on The Distribution of Low Molecular Weight Metabo- 2 lites in Mango Fruit by Air Flow Assisted Ionization-Mass 3
Spectrometry Imaging” and in my opinion manuscript can be accepted after minor revision
My comments are following
· Please improve abstract section
· From which region, author collected mango?
· Please improve conclusion section
· It is suggested to mention ultimate analysis of mango
Round 2
Reviewer 1 Report
The authors replied to the points I mentioned before.